# Joint Association of Dietary Protein Intake and Eating Habits with the Risk of Gestational Diabetes Mellitus: A Case-Control Study

**DOI:** 10.3390/nu15204332

**Published:** 2023-10-11

**Authors:** Kexin Gong, Lanci Xie, Yidan Cao, Xiayan Yu, Wenjing Qiang, Tuyan Fan, Tianli Zhu, Jingjing Liu, Fangbiao Tao, Beibei Zhu

**Affiliations:** 1Department of Maternal, Child and Adolescent Health, School of Public Health, Anhui Medical University, No 81 Meishan Road, Hefei 230032, China; 2145010455@stu.ahmu.edu.cn (K.G.); 2145010324@stu.ahmu.edu.cn (Y.C.); yuxiayan711@163.com (X.Y.); wenjingqiang0505@163.com (W.Q.); fty1731859326@163.com (T.F.); z1783546372@163.com (T.Z.); liujing05042022@163.com (J.L.); fbtao@ahmu.edu.cn (F.T.); 2Key Laboratory of Population Health Across Life Cycle, Anhui Medical University, Ministry of Education of the People’s Republic of China, No 81 Meishan Road, Hefei 230032, China; 3Ma’anshan Maternal and Child Health Care Hospital, Ma’anshan 243011, China; xielc77@sina.com; 4Anhui Provincial Key Laboratory of Population Health and Aristogenics, Anhui Medical University, No 81 Meishan Road, Hefei 230032, China; 5NHC Key Laboratory of Study on Abnormal Gametes and Reproductive Tract, No 81 Meishan Road, Hefei 230032, China

**Keywords:** dietary protein, eating habits, gestational diabetes mellitus, case-control study

## Abstract

Because the associations between different dietary protein sources and the risks of gestational diabetes mellitus (GDM) are inconsistent, and those of eating habits with GDM have rarely been explored, we aimed to investigate the independent and joint association of major dietary protein sources and eating habits with GDM in a case-control study including 353 GDM cases and 718 controls in China. Dietary protein intake and eating habits prior to GDM diagnosis were collected through questionnaires at 24~28 gestational weeks. Multivariate logistic regression was used to evaluate the independent and joint associations of dietary protein intake and eating habits with GDM. The Anderson model was used assess if there is an additive interaction between them. Animal protein, red meat protein and dairy products protein intake were significantly and positively associated with GDM. Among the eating habits, preferences for hot food, firm food and soft food were significantly associated with higher odds of GDM. Individuals with unhealthy eating habits and high dietary protein simultaneously had the highest odds of GDM, and the ORs were 2.06 (1.25, 3.41) for the total protein, 2.97 (1.78, 4.96) for animal meat, 3.98 (2.41, 6.57) for the red meat protein and 2.82 (1.81, 4.41) for the dairy protein; the *p* values for the trend were all significant (*p* < 0.001). However, no additive interaction was detected. In conclusion, our study found that dietary protein intake and eating habits prior to GDM diagnosis were both independently and jointly associated with the odds of GDM.

## 1. Introduction

Gestational diabetes mellitus (GDM) is defined as any degree of glucose intolerance with an onset or first recognition during pregnancy [1]. As of 2021, the global prevalence of GDM was as high as 16.7% [2]. There is a consensus that GDM causes immediate or long-term adverse health effects towards mothers and their offspring [3]. In particular, women with GDM are sevenfold more likely to develop type 2 diabetes mellitus (T2DM) later in life, and diabetes accounts for 12.2% of all causes of death [4]. Hence, it is crucial to prevent GDM through modifiable risk factors such as dietary intake.

Dietary protein is crucial for human health because it provides essential amino acids that act on the metabolic targets involved in satiety, energy expenditure and the sparing of fat-free mass [5]. Accordingly, many people consider the protein content when purchasing food and report trying to eat more protein. The global market for protein ingredients reached approximately USD 90 billion by 2021, largely driven by the growing demand for protein-fortified food products [6]. However, the growing intake of protein has gradually gained possible adverse health concerns, among which its role in the modulation of insulin resistance is a vital topic, and dietary protein was shown to act as a gluconeogenic precursor, thereby stimulating hexosamine biosynthesis or activating the mTOR-signaling pathway [7,8]. Thus, many previous studies have explored the associations between dietary protein intake and the risks of T2DM in the general population [9] and that of dietary protein intake prior to or during pregnancy and the risks of GDM [10,11,12,13], but different sources might have different effects. For example, most studies found that higher animal protein intake was associated with a higher T2DM [9] and GDM risk [10,11,12,13]; however, for plant protein, more diverse results were yielded [9,10,11,12], from a lower risk [9,10] to a null association [12] and increased risk [11]. Thus, more studies—in particular, those with more detailed protein sources—are needed to better clarify the precise associations between protein intake and GDM.

Eating habits refer to people’s preferences for foods, including dietary materials, cooking and eating speeds and taste [14]. They were suggested to relate to multiple chronic diseases [15,16,17,18,19], such as adiposity, metabolic syndrome [15], irritable bowel syndrome (IBS) [16], inflammatory bowel disease (IBD) [17], chronic constipation [18] and functional dyspepsia [19]. However, its association with GDM has been inadequately researched. To our knowledge, only one recent study conducted in Japan [20] found that, compared with women reporting a slow eating speed, those who reported a very fast eating speed had an increased incidence of GDM. This study only paid attention to the eating speed, while other dimensions have not been explored. Furthermore, the joint association between dietary protein and eating habits with GDM remains unclear. Therefore, based on a case-control study with retrospectively collected dietary data, our study aimed to examine (1) the independent associations between dietary protein, especially different sources, three eating habits (eating speed, food firmness preference, temperature) and the risk of GDM; (2) the joint association of major dietary protein sources and eating habits with GDM; and (3) if there was an interaction between them.

## 2. Methods

### 2.1. Study Participants and Design

Our current study was based on a case-control study conducted from June 2021 to September 2022 at the Ma’anshan Maternal and Child Health Care Center, Anhui Province, China. At 24~28 weeks of gestation, women who were aged ≥ 18 years, had no communication problems, were residents of Ma’anshan and had a singleton pregnancy were invited to participate. Based on the results of a 75 g oral-glucose-tolerance test (OGTT), women who met the GDM diagnosis criteria were included as the case group, and at the ratio of 1:2, women who were not diagnosed with GDM were included as the control group. The exclusion criteria were as follows: pre-pregnancy diabetes and previous pregnancy with hypothyroidism. A total of 1110 pregnant women were invited; after excluding 2 women with pre-pregnancy diabetes and 3 women with a previous pregnancy with hypothyroidism, 1105 women remained, of which 353 were GDM cases and 752 were controls. For our current study, we further excluded 34 women without dietary information, leaving a total of 1071 pregnant women, of which 353 were GDM cases and 718 were controls.

The details of the subject recruitment are shown in Appendix A. The study was approved by the Ethics Committee of Anhui Medical University (no. 20210732). All procedures in this study were performed according to the relevant guidelines and regulations, and written informed consent was obtained from each participant.

### 2.2. OGTT and Diagnosis of GDM

GDM was diagnosed after the 75 g OGTT if one or more of the following criteria were met: fasting plasma glucose (FPG) ≥ 5.1 mmol/L, 1 h plasma glucose (PG-1 h) ≥ 10.0 mmol/L or 2 h plasma glucose (PG-2 h) ≥ 8.5 mmol/L [21].

### 2.3. Maternal Dietary Assessment

Dietary information was collected using an interviewer-administrated semi-quantitative food frequency questionnaire (FFQ). This FFQ has been validated in Chinese pregnant women [22]. It consists of three parts: the food group, food intake frequency and food intake per visit, respectively. Meanwhile, this FFQ questionnaire has good reliability and validity. The intraclass correlation coefficients of FFQ for foods ranged from 0.23 to 0.49, and for nutrients, it ranged from 0.24 to 0.58. The energy-adjusted and de-attenuated correlation coefficients for foods ranged from 0.35 to 0.56, and for nutrients, it ranged from 0.11 to 0.63. The mean of the Pearson’s correlation coefficients for various foods and nutrients was 0.43. All trainers were uniformly trained to use standardized and normative language for questioning to ensure that the data obtained were as representative of the true level as possible. The participants were asked to recall their one-year-prior dietary intake of 11 major food groups, such as grains, beans, vegetables, fungi and algae, fruits, dairy and dairy products, meat, fish and shrimp, snacks and beverages. The frequency of consumption of each food item (annually, monthly, weekly, daily or not), the number of servings and the average amount per serving (g or ml) were collected. Based on each food item, the total protein, plant protein, animal protein and different sources of protein intake of each individual were calculated. The frequency and amount of consumption of each food or beverage per unit of time were converted into food consumption per day. Intakes of individual nutrients including protein were computed by multiplying the frequency of consumption of each unit of food by the nutrient content of the specified portions based on food composition data from China Food Composition Second Edition [23].

### 2.4. Maternal Eating Habits

We chose daily and representative eating habits and identified three eating habits (eating temperature, firmness and eating speed) that have been reported to be associated with other diseases, and we hypothesized they could also be relevant to GDM. Therefore, while surveying the dietary survey, each woman self-reported three items regarding eating habits, including eating speed (fast, moderate and slow), temperature preference (hot, moderate and cold) and food firmness preference (firm, moderate and soft).

### 2.5. Covariates

Demographic characteristic covariates were obtained through a face-to-face questionnaire used in a standardized interview for pregnant women at enrollment, including maternal age, education (middle school and below, high school or technical secondary college or junior college or regular college), average family income (<CNY 50,000, 50,000~99,900, 100,000~199,900 or ≥200,000/year), marital status (married, unmarried or other), age of menarche (8~11, 12, 13, 14 or ≥15), working status (brain-based work, physical-based work or currently not working), smoking (no (never smoked)/yes (smoking is defined as having smoked at least 100 cigarettes)), drinking (more than one time per month (340 mL of beer or 140 mL of wine or 43 mL of liquor), conception season (winter (December, January and February), spring (March, April and May), summer (June, July and August) or fall (September, October and November)) and sleep quality (excellent, good or average). The International Physical Activity Questionnaire [24] was used to evaluate moderate physical activity (dividing low, medium and high), adverse maternal history (spontaneous abortions, stillbirths, birth defects, etc.), family history of diabetes and family history of cardiovascular disease (immediate family members with diabetes or cardiovascular disease, typically parents). Maternal pre-pregnancy body mass index (BMI) was calculated using pre-pregnancy weight (kg) divided by height^2^ (m^2^).

### 2.6. Statistical Analysis

Descriptive variables were all approximately normally distributed and are reported as means ± SDs, and categorical variables were summarized as frequencies and percentages. Comparisons of basic characteristics and other food intake between different total protein intake groups were performed using chi-square tests for categorical variables and ANOVA or t tests for continuous variables with a normal distribution.

The protein intake from red meat, poultry, seafood, eggs and beans was divided into quartiles, in which quartile one was set as the reference group, and logistic regression analyses were used to evaluate the association between protein (including different sources of protein) intake and GDM; the results were presented as odds ratios (ORs) with a 95% confidence interval (CIs). The significance of linear trends across categories of protein intake was tested by assigning the median value for each quartile and analyzing this variable as a continuous variable in multivariate models. Covariates related to protein intake or GDM were chosen as potential confounders in multivariable analyses. The multivariate models were as follows: model one was a crude model; model two was adjusted for maternal age, pre-pregnancy BMI, family history of diabetes mellitus, physical activity, three eating habits, sleep quality, total calories, carbohydrate, total fat and cholesterol.

For eating habits, logistic regression was applied to examine the associations between individual eating habits (i.e., food temperature (hot, moderate and cold), firmness (firm, moderate and soft), eating speed (fast, moderate and slow)) and the odds of GDM, with the moderate category as a reference; the ORs and 95% CIs of GDM for the other two categories were calculated. Each eating habit category identified as increasing the odds of GDM was referred to as an unhealthy eating habit and assigned one point, and the total risk score for each individual was calculated by counting the number of risk items. It is worth noticing that the preferences for soft and firm food were exclusive. One can only choose either soft or firm but cannot choose both of them; thus, the highest score was two. The final risk scores of eating habits were assigned as zero to two points, higher meaning unhealthier. Using zero points as reference, the ORs and 95% CIs of GDM for the other two categories were calculated using multivariate logistic regression. The fully adjusted model included covariates such as age, pre-pregnancy BMI, family history of diabetes, physical activity, sleep quality, total calories, carbohydrate, total fat and cholesterol.

Furthermore, the joint association of protein intake and eating habits was assessed by calculating the odds of GDM for each combination using multivariate logistic regression; for protein intake, Q2~Q4 was regarded as a high intake and Q1 as a low intake, and for eating habits, the risk score point of zero was regarded as healthy, and that of one and above was regarded as unhealthy. Lastly, an additive interaction analysis of the protein intake and the eating habits regarding the odds of GDM was performed. The additive interaction was considered to be significant if any of the following items met the criteria: a 95% CI of relative excess risk due to interaction (RERI) and an attributable proportion due to interaction (AP) do not contain zero, or the 95% CI of the synergy index (S) does not contain one [25]. All statistical analyses were conducted using SPSS version 25.0 software (IBM Corp., Armonk, NY, USA), and two-sided *p* < 0.05 was regarded as being of statistical significance.

## 3. Results

### 3.1. Characteristics of the Study Population

For the 1071 pregnant women included in this study, the mean age was 30.90 ± 3.93 years, and the mean pre-pregnancy BMI was 22.42 ± 3.82 kg/m^2^. Compared with the controls, women with GDM were more likely to have a family history of diabetes, lower physical activity and an average sleep quality (Appendix A). Compared with the participants with a lower total protein intake, those with a higher protein intake were more likely to have higher education levels, a higher consumption of animal protein and plant protein and a higher intake of total calories, carbohydrates, total fat and cholesterol (Table 1 and Appendix A).

### 3.2. Associations between Dietary Protein Sources and Odds of GDM

The major protein sources, total protein, animal protein and eggs and dairy products were found to be significantly different between the control and GDM groups. In addition, they were all higher in the GDM group than in the control group (Table 2).

In logistic regression analyses, total dietary protein intake, animal protein intake, red meat intake and dairy products intake were significantly and positively associated with the odds of GDM in model one; the ORs (95% CI) were 1.52 (1.06, 2.19), 1.70 (1.17, 2.48), 2.35 (1.60, 3.46) and 1.64 (1.15, 2.33), respectively. Other sources of protein, including plant protein and eggs protein, were revealed to not be associated with the odds of GDM. However, in model two, when adjusted for age, pre-pregnancy BMI and family history of diabetes, physical activity, sleep quality, three eating habits, total calories, carbohydrate, total fat and cholesterol, only animal protein, red meat protein and dairy protein intake were significantly associated with the odds of GDM. Compared with the lowest quartile, the ORs (95% CI) for GDM of the second, third and highest quartile of the animal protein intake were 1.70 (1.04, 2.76), 2.14 (1.19, 3.88) and 1.68 (0.85, 3.29) (*p* for trend = 0.403), they were 2.12 (1.36, 3.29), 1.87 (1.22, 2.86) and 2.47 (1.57, 3.89) for the red meat protein intake (*p* for trend = 0.001) and they were 1.45 (0.98, 2.14), 1.49 (0.94, 2.36) and 1.64 (1.04, 2.57) for the dairy protein intake (*p* for trend = 0.019) (Figure 1).

### 3.3. Eating Habits with the Odds of GDM

We found that people who preferred hot food, firm food and soft food were significantly more prevalent in the GDM group than in the control group (Appendix A). Meanwhile, in the fully adjusted model, compared with preferences for a moderate food temperature and moderate firmness, preferences for hot food (OR (95% CI): 1.72 (1.12, 2.66)), firm food (OR (95% CI): 1.71 (1.17, 2.51)) and soft food (OR (95% CI): 1.47 (1.04, 2.15)) were significantly associated with increased odds of GDM (Table 3). Eating fast was shown to be significantly associated with increased odds of GDM; however, when further adjusting for the other two eating habits, the association was attenuated to insignificant. Further scoring of eating habits revealed that individuals with unhealthy eating habits were more prevalent in the GDM group (47.1%) than in the control group (32.3%). Compared with individuals with zero points, the ORs (95% CI) of GDM were 1.73 (95% CI: 1.29, 2.33) for one point and 2.88 (95% CI: 1.65, 5.04) for two points (Table 4).

### 3.4. The Joint Association of Dietary Protein and Eating Habits with the Odds of GDM

In the fully adjusted model, using the combination of healthy eating habits and low protein intake as a reference, for the total protein and animal protein, only the last combination (unhealthy eating habits and high protein intake) were associated with increased odds of GDM, and the ORs were 2.06 (1.25, 3.41) for total protein and 2.97 (1.78, 4.96) for animal protein. The additive interaction between a high total protein intake and unhealthy eating habits for GDM was insignificant (RERI: −0.02 (95% CI, −1.11 to 1.07)) and was also insignificant between a high animal protein intake and unhealthy eating habits for GDM (RERI: −0.33 (95% CI, −1.98 to 1.32)). For the red meat protein, a higher intake combined with eating habits, healthy (OR = 2.11, 95% CI = 1.29, 3.45) or not (OR = 3.98, 95% CI = 2.41, 6.57), was significantly associated with increased odds of GDM; however, a insignificant additive interaction was detected (RERI: 0.70 (95% CI, 1.40 to 2.79)). For dairy protein, all of the three other combinations were associated with significantly increased odds of GDM, with ORs (95% CI) of 2.03 (1.19, 3.47), 1.55 (1.01, 2.37) and 2.82 (1.81, 4.41), respectively. The *p* values for the trend were all significant (*p* < 0.001). However, no additive interaction was observed between dairy protein intake and eating habits regarding the odds of GDM (RERI: 0.62 (95% CI, −1.61 to 1.85)) (Figure 2).

## 4. Discussion

Our study found that higher intakes of animal protein, red meat and dairy protein were associated with higher odds of GDM in a Chinese population, whereas protein from other sources had no significant associations with GDM. More importantly, for the very first time, we found that, aside from eating hot food, preferences for firm food and soft food were both associated with increased odds of GDM as well. Notably, individuals with a higher intake of (total, animal, red meat, dairy) protein combined with unhealthy eating habits were found to have the highest odds of GDM; however, no additive interaction was observed between them.

In the unadjusted model and the model only adjusted for age, pre-pregnancy BMI and family history of diabetes, total protein intake in our study was found to be positively associated with the odds of GDM, which was generally consistent with prior studies [10,11,12]. However, when further adjusted for physical activity, sleep quality, three eating habits, total calories, carbohydrate, total fat and cholesterol, the association was attenuated to insignificant. This change in the results indicated that total protein intake might not be independently associated with the odds of GDM but confounded by other factors, and different sources of protein may have divergent effects on GDM, which was supported by our further analysis. When total protein was divided into animal and plant sources, compared with the lowest quartile of animal protein, significant associations were observed for the second and third quartiles; however, they were not observed for the highest quartile. Most of the previous studies suggested that animal protein intake was associated with an increased risk of GDM [10,11,12,13]. One explanation for this disagreement is that the average animal protein intake of our population was considerably lower, with median protein intakes of 1.7% and 4.0% of energy in the lowest and highest quartiles, respectively, compared with 10.0% and 18.6% of energy in the lowest and highest quartiles in the Nurses’ Health Study II; another reason might be that we considered the confounding of eating habits. For plant protein, our results showed no significant association between plant protein intake and GDM, and a further exploration of the plant protein sources still did not reveal any particular plant protein associated with the odds of GDM. However, controversial results existed in previous studies, ranging from inverse association [10] to null association [12] and even positive association [11]. The discrepancies might due to different dietary patterns, different sources, the types of plant protein or residual confounding.

Amino acids were revealed as key signaling molecules [26] in explaining how protein intake influences metabolism. However, the amino acid composition is considered to be very different between plant and animal proteins [27,28]. For example, branched-chain amino acids (BCAAs), a class of amino acids more abundant in proteins of animal origin, have been reported to interfere with insulin signaling via the stimulation of mTOR and S6K1 and the phosphorylation of IRS1 on serine residues, thus potentially affecting diabetes-related metabolic pathways [29,30,31]. In contrast, certain amino acids rich in proteins of plant origin, such as arginine, were thought to have a beneficial effect on the body’s insulin metabolism [32,33]. Thus, the diverse effects of different compounds in animal and plant protein might partly explain those differences. When subdividing animal proteins, red meat and dairy products proteins were shown to be significantly associated with higher odds of GDM. Regarding red meat, its associations with T2DM and GDM have been well established in previous studies [10,12,34]. In addition to protein from red meat, we found that a higher dairy protein intake was also associated with higher odds of GDM, which is consistent with the GUSTO cohort study [11]. Dairy protein in our study included milk and yogurt, and only the intake of milk protein was found to be significantly higher in the GDM group, but not yogurt (Appendix A). Our results were inconsistent with most of the previous studies, in which low-fat milk consumption was generally associated with a lower risk of T2DM [35,36], and high-fat milk had no association with the risk of GDM [36,37]. However, there was a study showing that people who drank whole milk twice a week or more had a higher risk of T2DM than those who drank it less than once a month [35]. Thus, further research is needed to examine whether the association between milk and GDM risk is due to dairy types, other components of dairy, genetic backgrounds, etc.

More importantly, three eating habit preferences (soft food, firm food and hot food) were observed to be significantly associated with GDM. To our knowledge, only one study from Japan investigated the association between eating habits and the risk of GDM; however, this study only paid attention to eating speed, indicating that a fast eating speed increased the risk of GDM. Originally, our results also suggested that a fast eating speed increased the odds of GDM; however, when further adjusted for the other two eating habits, the association was attenuated to insignificant, which indicated that eating fast might not be independently associated with the odds of GDM but rather confounded by other eating habits—for example, food firmness preference and food temperature.

Interestingly, preferences for both soft food and firm food were associated with higher odds of GDM. Regarding a preference for soft food, its influence on glucose metabolism has been well demonstrated in previous studies—for example, a randomized crossover trial of patients with type 2 diabetes [38] showed that starch pasting in the form of cooked porridge had a more significant effect on postprandial blood glucose elevation than the consumption of natural starch in the uncooked form. A possible explanation for this may be that heating with water makes the starch granules in rice swell and starchify [39], which can cause them to come in contact with digestive juices in large quantities and be more easily absorbed. In addition, heating with water also leads to starch structure transformation, from starch hydrolysis into dextrin or maltose [40], both of which are easily hydrolyzed in the digestive tract, hydrolyzed to glucose and absorbed rapidly [41]; thus, a preference for eating soft food could lead to a rapid post-prandial blood glucose increase, imposing a burden on the pancreatic islet [38]. Oddly, a preference for firm food was revealed to be associated with increased odds of GDM as well in our study, which has never been reported before; thus, the underlying mechanism was unclear and needs further exploration. In our study, we also found that eating hot food was independently associated with increased odds of GDM. No direct evidence of an association between blood glucose and food temperature in population and animal studies was found, and the most similar association was found in the study on food temperature and esophageal cancer [42,43,44]. A meta-analysis found that the consumption of food and beverages of a high enough temperature is associated with an increased risk of esophageal cancer, particularly ESCC [42]. This is due to several relevant observations. On the one hand, eating hot food causes localized inflammation of the tract tissues. Inflammation leads to the production of endogenous reactive nitrogen, including nitrosamines. On the other hand, changes in eating habits lead to repeated irritation of the esophageal mucosa or changes in the esophageal microbiota. It has even been found that a higher incidence of p53 mutations (G:C to A:T mutations at the CpG locus) has been found in esophageal tumors in patients who have been eating hot foods for a long time [44].

Notably, we further found that individuals with a high protein intake (total protein, animal protein, red meat or dairy products) combined with unhealthy eating habits have the highest odds of GDM. Although, previously, total protein had no significant association with GDM solely, when combined with unhealthy eating habits, individuals with higher total protein were also shown to have increased odds of GDM. Our study indicated that eating habits are important both in sole and joint ways, and they have long been neglected.

Our study has several advantages. First, this is the first study to evaluate comprehensive eating habits with the odds GDM, which adds new clues for GDM etiology. Second, a large number of confounders were adjusted for in this study—especially, eating habits and dietary protein intake were mutually adjusted, making our results more reliable. Third, we examined the joint association of dietary protein and eating habits with GDM, which could provide evidence for more precise dietary recommendations for GDM prevention.

However, several limitations should be noted as well. First, we cannot fully rule out recall bias and reverse causation, for the dietary information was collected after women noticed the OGTT results, and they recalled information one year prior to the survey. Second, in addition to the potential confounders, we included in the analysis other factors that could also influence GDM and diet, which might interfere with our findings. For example, the cooking methods, fasting glucose level and serum uric acid level in normal or abnormal ranges cannot be completely ruled out. Third, our study did not distinguish between different periods—for example, pre-pregnancy, early pregnancy or mid-pregnancy—which made it impossible to identify the critical time window of the impact of dietary protein intake on the risk of GDM. Finally, differences in the diets of people in different regions limit the generalizability of our findings.

## 5. Conclusions

In our case-control study, we found that preferences for hot, soft and firm food and a higher intake of dietary protein (animal, red meat, dairy) were associated with higher odds of GDM; when the two factors were combined, the odds were further elevated, which suggests both independent and joint associations of unhealthy eating habits and dietary protein intake with GDM.

## Figures and Tables

**Figure 1 nutrients-15-04332-f001:**
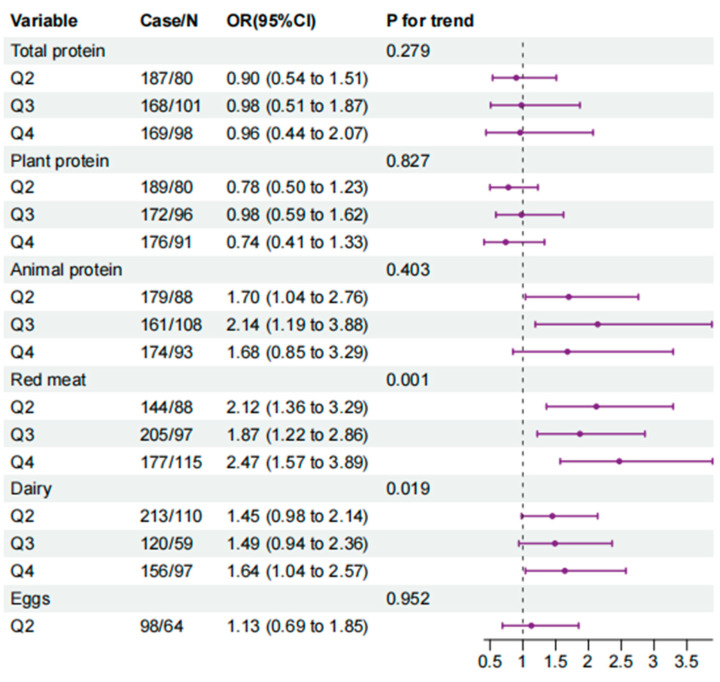
Forest plot of the association between protein intake and GDM. The model is adjusted for age, pre-pregnancy BMI, family history of diabetes, physical activity, sleep quality, three eating habits, total calories, carbohydrate, total fat and cholesterol. The *p* value for the trend is based on the variable containing the median value for each quintile.

**Figure 2 nutrients-15-04332-f002:**
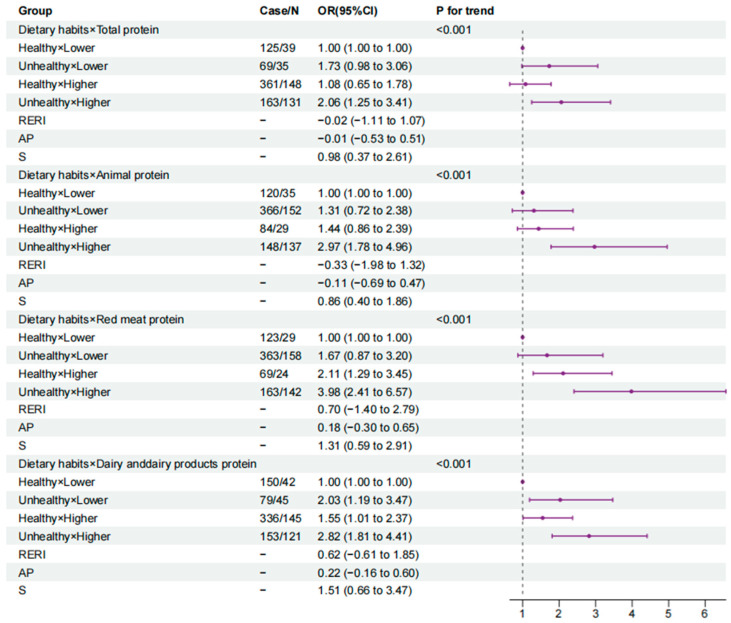
Forest plot of combined effects of eating habits and protein intake on GDM (0 Health/1~2 Unhealthy; Q1 Low/Q2~Q4 High). Abbreviations: AP, attributable proportion due to interaction; RERI, relative excess risk due to interaction; SI, synergy index. Model is adjusted for age, pre-pregnancy BMI, family history of diabetes, physical activity, sleep quality, total calories, carbohydrate, total fat and cholesterol.

**Table 1 nutrients-15-04332-t001:** Baseline characteristics of pregnancy according to the quartiles of dietary protein intake ^a^ (*n*(%)).

Characteristic	Total (*n* = 1071)	Protein Intake Quartiles				*p*-Value ^b^
		Q1 (*n* = 268)	Q2 (*n* = 267)	Q3 (*n* = 269)	Q4 (*n* = 267)	
Age (years)	30.90 ± 3.93	30.37 ± 4.06	30.99 ± 3.63	31.06 ± 4.00	31.16 ± 4.02	0.086
Pre-pregnancy BMI	22.42 ± 3.82	22.85 ± 3.79	22.25 ± 3.69	22.13 ± 3.57	22.46 ± 4.18	0.140
Ethnicity						0.369
Han	1052 (98.2)	260 (97.0)	264 (98.9)	265 (98.5)	263 (98.5)	
Other	19 (1.8)	8 (3.0)	3 (1.1)	4 (1.5)	4 (1.5)	
Marital Status						0.193
Married	1059 (98.9)	264 (98.5)	267 (100.0)	266 (98.9)	262 (98.1)	
Unmarried or other	12 (1.1)	4 (1.5)	0 (0.0)	3 (1.1)	5 (1.9)	
Annual household income (CNY)						0.190
<50,000	159 (14.9)	50 (18.7)	33 (12.4)	37 (13.8)	39 (14.6)	
50,000~99,900	369 (34.5)	99 (37.1)	96 (36.0)	81 (30.1)	93 (34.8)	
100,000~199,900	387 (36.2)	90 (33.7)	93 (34.8)	107 (39.8)	97 (36.3)	
≥200,000	155 (14.5)	28 (10.5)	45 (16.9)	44 (16.4)	38 (14.2)	
Education status						0.826
Middle school and below	176 (16.4)	50 (18.7)	46 (17.2)	42 (15.6)	38 (14.2)	
High school or technical secondary college	223 (20.8)	58 (21.6)	52 (19.5)	58 (21.6)	55 (20.6)	
Junior college or regular college	672 (62.7)	160 (59.7)	169 (63.3)	169 (62.8)	174 (65.2)	
Working Status						0.296
Brain-based work	585 (54.6)	134 (50.0)	159 (59.6)	144 (53.5)	148 (55.4)	
Physical-based work	69 (6.4)	18 (6.7)	13 (4.9)	16 (5.9)	22 (8.2)	
Currently not working	417 (38.9)	116 (43.3)	95 (35.6)	109 (40.5)	97 (36.3)	
Smoking						0.266
Yes	40 (3.7)	14 (5.2)	7 (2.6)	7 (2.6)	12 (4.5)	
No	1031 (96.3)	254 (94.8)	260 (97.4)	262 (97.4)	255 (95.5)	
Drinking						0.024
Yes	172 (16.1)	58 (21.6)	42 (15.7)	39 (14.5)	33 (12.4)	
No	899 (83.9)	210 (78.4)	225 (84.3)	230 (85.5)	234 (87.6)	
Age of menarche (years)						0.801
8~11	33 (3.1)	7 (2.6)	13 (4.9)	8 (3.0)	5 (1.9)	
12	207 (19.3)	58 (21.6)	48 (18.0)	53 (19.7)	48 (18.0)	
13	289 (27.0)	68 (25.4)	76 (28.5)	68 (25.3)	77 (28.8)	
14	257 (24.0)	64 (23.9)	64 (24.0)	62 (23.0)	67 (25.1)	
≥15	285 (26.6)	71 (26.5)	66 (24.7)	78 (29.0)	70 (26.2)	
Conception Season						<0.001
Spring	287 (26.8)	92 (34.3)	84 (31.5)	71 (26.4)	40 (15.0)	
Autumn	166 (15.5)	53 (19.8)	50 (18.7)	41 (15.2)	22 (8.2)	
Summer	154 (14.4)	30 (11.2)	31 (11.6)	34 (12.6)	59 (22.1)	
Winter	464 (43.3)	93 (34.7)	102 (38.2)	123 (45.7)	146 (54.7)	
Adverse maternal history						0.069
No	922 (86.1)	242 (90.3)	230 (86.1)	230 (85.5)	220 (82.4)	
Yes	149 (13.9)	26 (9.7)	37 (13.9)	39 (14.5)	47 (17.6)	
Family history of diabetes						0.236
No	792 (73.9)	195 (72.8)	209 (78.3)	192 (71.4)	196 (73.4)	
Yes	228 (21.3)	56 (20.9)	50 (18.7)	60 (22.3)	62 (23.2)	
Unclear	51 (4.8)	17 (6.3)	8 (3.0)	17 (6.3)	9 (3.4)	
Family history of cardiovascular disease						0.656
No	833 (77.8)	209 (78.0)	201 (75.3)	218 (81.0)	205 (76.8)	
Yes	175 (16.3)	41 (15.3)	50 (18.7)	36 (13.4)	48 (18.0)	
Unclear	63 (5.9)	18 (6.7)	16 (6.0)	15 (5.6)	14 (5.2)	
Physical activity						0.007
Low	211 (19.7)	52 (19.4)	59 (22.1)	52 (19.3)	48 (18.0)	
Medium	562 (52.5)	119 (44.4)	146 (54.7)	140 (52.0)	157 (58.8)	
High	298 (27.8)	97 (36.2)	62 (23.2)	77 (28.6)	62 (23.2)	
Sleep quality						0.141
Excellent	19 (1.8)	2 (0.7)	6 (2.2)	6 (2.2)	5 (1.9)	
Good	129 (12.0)	34 (12.7)	37 (13.9)	38 (14.1)	20 (7.5)	
Average	923 (86.2)	232 (86.6)	224 (83.6)	225 (83.6)	242 (90.6)	

^a^ Continuous variables are presented as the mean ± SD; categorical variables are presented as the *n* (%). BMI, body mass index. ^b^ *p*-values are from ANOVA for continuous data and from chi-square tests for categorical data.

**Table 2 nutrients-15-04332-t002:** Major dietary protein source intakes in different groups ($\bar{X}\pm S$).

Nutrients	Total	GDM	Control	*p* Value ^a^
	(*n* = 1071)	(*n* = 353)	(*n* = 718)	
Protein, g/day				
Total protein	71.13 ± 26.66	73.52 ± 25.95	69.95 ± 26.94	0.040
Plant protein	30.65 ± 28.22	30.78 ± 13.75	30.59 ± 14.17	0.831
Animal protein	40.48 ± 20.29	42.74 ± 20.52	39.37 ± 20.10	0.011
Protein sources				
Plant protein, g/day				
From grains	21.43 ± 20.22	20.99 ± 8.55	21.65 ± 10.46	0.304
From beans	6.42 ± 4.00	6.95 ± 9.81	6.15 ± 7.87	0.151
From nuts	1.42 ± 1.03	1.44 ± 1.53	1.42 ± 1.56	0.840
From soy milk	1.38 ± 1.03	1.40 ± 1.67	1.37 ± 1.58	0.734
Animal protein, g/day				
From red meat	12.01 ± 11.50	12.90 ± 11.04	11.57 ± 11.70	0.075
From poultry	5.21 ± 2.86	5.24 ± 8.61	5.19 ± 7.53	0.916
From eggs	6.54 ± 6.65	6.95 ± 3.55	6.34 ± 4.04	0.017
From dairy products	7.41 ± 7.50	7.99 ± 4.58	7.13 ± 4.50	0.004
From fish	5.08 ± 2.53	5.26 ± 6.82	4.98 ± 6.56	0.519
From shrimp	4.25 ± 2.60	4.42 ± 4.92	4.16 ± 5.04	0.424

^a^ *p*-values are from ANOVA.

**Table 3 nutrients-15-04332-t003:** Associations of eating habits with the odds of GDM (OR (95% CI)).

Eating Habits	GDM (*n* = 353)	Model One	Model Two	Model Three
Food temperature				
Moderate	272 (77.1)	1.00	1.00	1.00
Hot	54 (15.3)	2.14 (1.44, 3.20)	2.16 (1.41, 3.31)	1.72 (1.12, 2.66)
Cold	27 (7.6)	1.03 (0.64, 1.67)	0.87 (0.52, 1.46)	070 (0.41, 1.18)
Firmness				
Moderate	209 (59.2)	1.00	1.00	1.00
Firm	67 (19.0)	1.78 (1.25, 2.54)	1.82 (1.25, 2.65)	1.71 (1.17, 2.51)
Soft	77 (21.8)	1.64 (1.18, 2.28)	1.63 (1.14, 2.33)	1.47 (1.04, 2.15)
Eating speed				
Moderate	227 (64.3)	1.00	1.00	1.00
Fast	90 (25.5)	1.46 (1.07, 1.99)	1.41 (1.01, 1.96)	1.24 (0.88, 1.74)
Slow	36 (10.2)	0.84 (0.56, 1.28)	0.96 (0.62, 1.49)	0.84 (0.54, 1.31)

Model one is unadjusted. Model two is adjusted for age, pre-pregnancy BMI, family history of diabetes, physical activity, sleep quality, total protein, total calories, carbohydrate, total fat and cholesterol. Model three is adjusted for age, pre-pregnancy BMI, family history of diabetes, physical activity, sleep quality, total protein, total calories, carbohydrate, total fat, cholesterol and three dietary habits adjusted to each other.

**Table 4 nutrients-15-04332-t004:** Eating habits scores and the odds of GDM (OR (95% CI)).

Group	Total (*n* = 1071)	GDM (*n* = 353)	Model One	Model Two	*p* for Trend
Cumulative score for three diabetes habits					<0.001
0	673 (62.8)	187 (53.0)	1.00	1.00	
1	335 (31.3)	134 (38.0)	1.81 (1.38, 2.38)	1.73 (1.29, 2.33)	
2	63 (5.9)	32 (9.1)	2.81 (1.67, 4.73)	2.88 (1.65, 5.04)	

Model one is unadjusted. Model two is adjusted for age, pre-pregnancy BMI, family history of diabetes, physical activity, sleep quality, total calories, carbohydrate, total fat and cholesterol.

## Data Availability

The data described in the manuscript, code book and analytic code will be made available upon request.

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
