# Peer review of "Joint Association of Dietary Protein Intake and Eating Habits with the Risk of Gestational Diabetes Mellitus: A Case-Control Study"

_nutrients, 2023, doi:10.3390/nu15204332_

Round 1
Reviewer 1 Report
nutrients-2613979
Joint association of dietary protein intake and eating habits with the risk of gestational diabetes mellitus: A case-control study
I think this is an interesting and relevant work that can provide relevant information to create specific dietary recommendations for pregnant women to try to avoid GDM. However, there are many gaps in the methodology that make this study not entirely transparent.
Abstract
1. You can remove the headings “background”, “methods”…
2. Add the objective
3. Why do you only present CIs - ORs for eating habits?
4. Why do you indicate that preferring soft or firm food is an unhealthy habit?
5. I think that without the objective of the study it is difficult to know which results to emphasise. I recommend clarifying the objective and focusing the abstract on the main results. If it is eating habits, you should explain how, which and why they are considered unhealthy.
Introduction
6. Lines 50-58: Specify if these results are in pregnant women.
Methods
7. Indicate the ethics committee that approved this study.
8. Indicate if this FFQ is validated in pregnant population
9. Lines 115-177: In my opinion, this information is not relevant as you did not interview pregnant-women’s friends.
10. A more detailed information on maternal eating habits should be provided. Is there a classification or a valid test? Why did you use these three?
11. In some part of the manuscript you indicate that it is retrospective. Please, explain this in study design.
12. Line 139: I understand that all your quantitative variables present normal distributions as you described them as mean SD. Specify.
13. Lines 157-159: I understand the scores, but why the higher the unhealthier? is there a consensus about that? Why usually eating cold and soft food is unhealthier?
Results
14. Table 1: specify what drinking means
15. Table 2: Is more interesting for the current manuscript to present the supplementary table 2 than the one in main text.
16. Table 3: Change “normal” for “control”
17. Figure 2: Describe RERI, AP, S
18. Line 199: Why do you mean with “incident GDM”. Indicate somewhere in the text how many new cases you detected.
Results/discussion
19. Throughout the article you talk about "an increased risk of GDM", however you have not studied the incidence of GDM, you have simply compared cases (prevalent) and controls. I do not think it is correct to use the term risk for a disease that is already present. In any case, association. In fact, in some parts of the methods I doubt that this paper is a case-control study, sometimes it seems a descriptive cross-sectional study in which you compared protein intake and eating behaviours between mothers with GDM and those without GDM. It is necessary a better explain of the study design (especially on the exposition how and when was recorded).
Limitations
20. It may be necessary to include the difficulty of generalising these results to pregnant women in other countries due to differences in diet.
Conclusion
21. The conclusion should give a more concrete and direct message in relation to the objective of the study.
Reviewer 2 Report
The manuscript raises a very interesting topic. It is prepared correctly. However, I have a few minor comments about the manuscript:
1. Line 43-59: I suggest moving those sentences to the discussion section.
2. Line 83: The authors list pregnancy complications as exclusion criteria from the study. Heart failure and severe anemia were specified. Are these the only pregnancy complications that were the exclusion criteria? Please specify
Reviewer 3 Report
I’ve read with attention the paper of Gong et al. that is potentially of interest. The background and aim of the study have been clearly defined. The methodology applied is overall correct, the results are reliable and adequately discussed. I’ve only some minor comments:
- "Eating habit" is not immediately clear. Does the authors mean "behaviours"?
- Among the limitations, the authors should aknowledged that they considered a large number of dummies variables, but very few continuous ones. In particular, the models should be adjusted by fasting glucose level. Serum uric acid level should be also considered a predictor of incident diabetes. This has to be clearly discussed.
The quality of English language is quite good, with some minor typos.
Round 2
Reviewer 1 Report
Please, see the attachment.

Reviewer 3 Report
The authors have considered the reviewers' suggestion and improved the paper accordingly. I've no further comments on it.
Author Response
Thank you for your comments.
Round 3
Reviewer 1 Report
Thank you for replying to all my comments.
Author Response
Thank you for your comments.